# PHASE CONDUCTOR ON MULTI-LAYERED ATTENTIONS FOR MACHINE COMPREHENSION

## ABSTRACT

Attention models have been intensively studied to improve NLP tasks such as machine comprehension via both question-aware passage attention model and self-matching attention model. Our research proposes *phase conductor* (PhaseCond) for attention models in two meaningful ways. First, PhaseCond, an architecture of multi-layered attention models, consists of multiple phases each implementing a stack of attention layers producing passage representations and a stack of inner or outer fusion layers regulating the information flow. Second, we extend and improve the dot-product attention function for PhaseCond by simultaneously encoding multiple question and passage embedding layers from different perspectives. We demonstrate the effectiveness of our proposed model PhaseCond on the SQuAD dataset, showing that our model significantly outperforms both state-of-the-art single-layered and multiple-layered attention models. We deepen our results with new findings via both detailed qualitative analysis and visualized examples showing the dynamic changes through multi-layered attention models.

## 1 INTRODUCTION

Attention-based neural networks have demonstrated success in a wide range of NLP tasks ranging from neural machine translation (Bahdanau et al., 2015), image captioning (Xu et al., 2015), and speech recognition (Chorowski et al., 2015). Benefiting from the availability of large-scale benchmark datasets such as SQuAD (Rajpurkar et al., 2016), the attention-based neural networks has spread to machine comprehension and question answering tasks to allow the model to attend over past output vectors (Wang & Jiang, 2017; Seo et al., 2017; Xiong et al., 2017; Hu et al., 2017; Wang et al., 2017; Pan et al., 2017). Wang & Jiang (2017) uses attention mechanism in Pointer Network to detect an answer boundary by predicting the start and the end indices in the passage. Seo et al. (2017) introduces a bi-directional attention flow network that attention models are decoupled from the recurrent neural networks. Xiong et al. (2017) employs a coattention mechanism that attends to the question and document together. Wang et al. (2017) uses a gated attention network that includes both question and passage match and self-matching attentions. Both Pan et al. (2017) and Hu et al. (2017) employs the structure of multi-hops or iterative aligner to repeatedly fuse the passage representation with the question representation as well as the passage representation itself.

Inspired by the above-mentioned works, we are proposing to introduce a general framework PhaseCond for the use of multiple attention layers. There are two motivations. First, previous research on the self-attention model is to purely capture long-distance dependencies (Vaswani et al., 2017), and therefore a multi-hops architecture (Hu et al., 2017; Pan et al., 2017) is used to alternatively captures question-aware passage representations and refines the results by using a self-attention model. In contrast to the multi-hops and interactive architecture, our motivation of using the self-attention model for machine comprehension is to propagate answer evidence which is derived from the preceding question-passage representation layers. This perspective leads to a different attention-based architecture containing two sequential phases, question-aware passage representation phase and evidence propagation phase.

Second, unlike the domains such as machine translation (Bahdanau et al., 2015) which jointly align and translate words, question-passage attention models for machine comprehension and question answering calculate the alignment matrix corresponding to all question and passage word pairs (Wang & Jiang, 2017; Seo et al., 2017). Despite the attention models' success on the machine comprehen-

Table 1: Comparison of attention architectures of competing approaches: BIDAF (Seo et al., 2017), RNET (Wang et al., 2017), MReader (Hu et al., 2017), and PhaseCond (our proposed model).

| Model | Q-P Attention | Self-Attention | Structure | Fusion |
|---|---|---|---|---|
| BIDAF | $W^T[H;U;H \circ U]$ | N/A | Single | Bi-LSTM |
| RNET | $V^T \tanh(W^T[H_{t-1,t};U])$ | $V^T \tanh(W^T[H;U])$ | Single | Gate |
| MReader | $\text{softmax}(HU^T)U$ | $\text{softmax}(HH^T)H$ | Alternative | Gate |
| PhaseCond | $\text{softmax}(HU^T)V$ | $\text{softmax}(HH^T)H$ | Phased, Stacking | Inner/Outer |

sion task, there has not been any other work exploring learning to encode multiple representations of question or passage from different perspectives for different parts of attention functions. More specifically, most approaches use two same question representations $U$ for the question-passage attention model $\alpha(H, U)U$, where $H$ is the passage representation. Our hypothesis is that attention models can be more effective by learning different encoders for a question representation $U$ and a question representation $V$ from different aspects. The key differences between our proposed model and competing approaches are summarized at Table 1.

Our contributions are threefold: 1) we proposed a *phase conductor* for attention models containing multiple phases, each with a stack of attention layers producing passage representations and a stack of inner or outer fusion layers regulating the information flow, 2) we present an improved attention function for question-passage attention based on two kinds of encoders: an independent question encoder and a weight-sharing encoder jointly considering the question and the passage, as opposed to most previous works which only using the same encoder for one attention model, and 3) we provide both detailed qualitative analysis and visualized examples showing the dynamic changes through multi-layered attention models. Experimental results show that our proposed PhaseCond lead to significant performance improvements over the state-of-the-art single-layered and multi-layered attention models. Moreover, we observe several meaningful trends: a) during the question-passage attention phase, repeatedly attending the passage with the same question representation "forces" each passage word to become increasingly closer to the original question representation, and therefore increasing the number of layers has a risk of degrading the network performance, b) during the self-attention phase, the self-attention's alignment weights of the second layer become noticeably "sharper" than the first layer, suggesting the importance of fully propagating evidence through the passage itself.

## 2  MODEL ARCHITECTURE

We proposed phased conductor model (or PhaseCond), which consisting of multiple phases and each phase has two parts, a stack of attention layers $\mathcal{L}$ and a stack of fusion layers $\mathcal{F}$ controlling information flow. In our model, a fusion layer $\mathcal{F}$ can be an inner fusion layer $\mathcal{F}_{inner}$ inside of a stack of attention layers, or an outer fusion layer $\mathcal{F}_{outer}$ immediately following a stack of attention layers. Without loss of generality, PhaseCond's configurable computational path for two-phase, a question-passage attention phase containing $N$ question-passage attention layers $\mathcal{L}^Q$, and a self-attention phase containing $K$ self-attention layers $\mathcal{L}^S$, can be defined as $\{\mathcal{L}^Q \rightarrow \mathcal{F}_{inner}\}_{\times N} \rightarrow \mathcal{F}_{outer} \rightarrow \{\mathcal{L}^S \rightarrow \mathcal{F}_{inner}\}_{\times L} \rightarrow \mathcal{F}_{outer}$.

Figure 1 gives an concrete example of building PhaseCond based network for the machine comprehension task. The network contains encoding layers, question-passage attention layers, self-attention layers and output layers. The encoding layer maps various groups of features, such as character features and word features, to their corresponding embeddings. Those raw embeddings are then fed into an outer fusion layer to encode these embeddings as passage or question representations in Section 2.1. Next, the representations are sent to question-passage attention layers to align and represent passage representation with the whole question representation in Section 2.3. The output of each layer is concatenated and regularized by a stack of fusion layers in Section 2.2.1. After that, the question-attended passage representation is directly matching against itself, for the purpose of propagating information through the whole passage detailed in Section 2.3. For each self-attention layer, we configure an inner fusion layer to obtain a gated representation that is learned to decide how much of the current output is fused by the input from the previous layer detailed in Section 2.3.1.

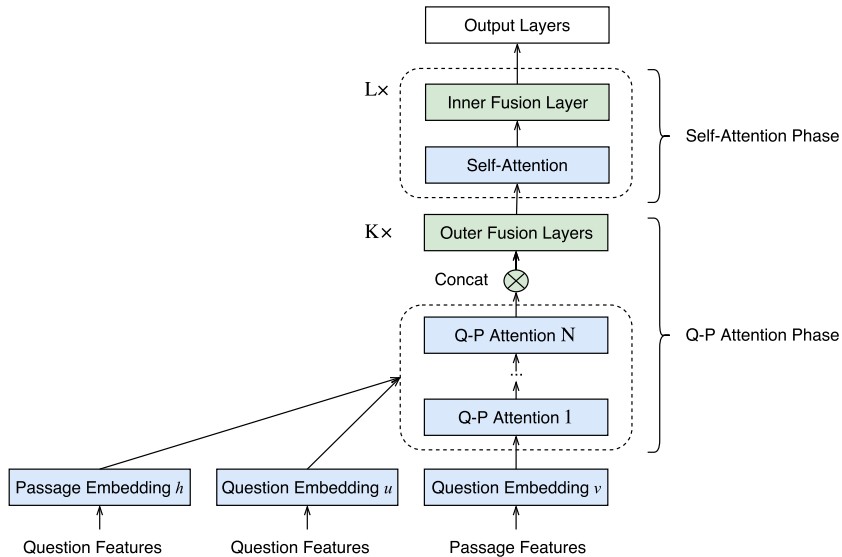

Figure 1: PhaseCond: our proposed attention model structure overview. We use the colored rectangle to highlight the focus of this paper. The question and passage encoder layers and attention layers are colored in blue, the fusion layers are colored in green.

Finally, the fused vectors are sent to the output layer to predict the boundary of the answer span described in Section 2.4.

## 2.1   ENCODER LAYERS

The concatenation of raw features as inputs are processed in fusion layers followed by encoder layers to form more abstract representations. Here we choose a bi-directional Long Short-Term Memory (LSTM) (Hochreiter & Schmidhuber, 1997) to obtain more abstract representations for words in passages and questions.

Different from the commonly used approaches that every single model has exactly one question and passage encoder (Seo et al., 2017; Wang et al., 2017; Hu et al., 2017), our encoder layers simultaneously calculate multiple question and passage representations, for the purpose of serving different parts of attention functions of different phases. We use two types of encoders, *independent encoder* and *shared encoder*. In terms of independent encoder, a bi-directional LSTM is used to produce new representation $v_1^Q, \ldots, v_m^Q$ of all words in the question,

$$v_j^Q = \text{BiLSTM}^Q(v_{j-1}^Q, v_j^Q) \tag{1}$$

where $v_j^Q \in \mathbb{R}^{2d}$ are concatenated hidden states of two independent BiLSTM for the $j$-th question word and $d$ is the hidden size.

In terms of shared encoder, we jointly produce new representation $h_1^P, \ldots, h_n^P$ and $u_1^Q, \ldots, u_m^Q$ for the passage and question via a shared bi-directional LSTM,

$$h_i^P = \text{BiLSTM}^S(h_{i-1}^P, h_i^P) \tag{2}$$

$$u_j^Q = \text{BiLSTM}^S(u_{j-1}^Q, u_j^Q) \tag{3}$$

where $h_i^P \in \mathbb{R}^{2d}$ and $u_j^Q \in \mathbb{R}^{2d}$ are concatenated hidden states of BiLSTM for the $i$-th passage word and $j$-th question word, sharing the same trainable BiLSTM parameters.

## 2.2   QUESTION-PASSAGE ATTENTION LAYERS

The process of representing a passage with a question essentially includes two sub-tasks: 1) calculating the similarity between the question and different parts of the passage, and 2) representing the passage part with the given question depending on how similar they are.

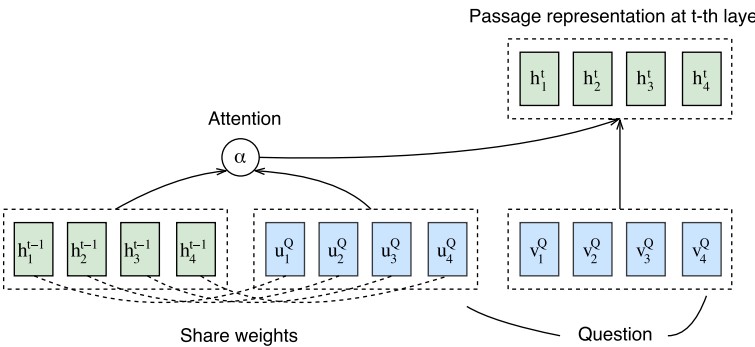

Figure 2: Improved question-passage attention model. We use blue color to denote question representations and use green color for passage representations.

A single question-passage attention layer is illustrated in Figure 2. In this model, at the $t$-th layer an alignment matrix $\mathcal{A}^t \in \mathbb{R}^m$, whose shape equals the number of words $n$ in a passage multiplied by the number of words $m$ in a question, is derived by aligning the passage representation at the $t-1$ layer with the shared weight question representation,

$$\mathcal{A}^t(i,j) = \frac{\exp(h_i^{t-1} \cdot u_j^Q)}{\sum_k \exp(h_i^{t-1} \cdot u_k^Q)} \tag{4}$$

where $h_i^{t-1}$ is the the $i$-th passage word representation at the $t-1$ layer, $h_i^0$ equals to $h_i^P$ calculated from Eq 2, $u_j^Q$ calculated from Eq 3 is the same for all the layers, the alignment matrix element $\mathcal{A}^t(i,j)$ is a scalar, denoting the similarity between the $i$-th passage word and the $j$-th question word by using dot product of the passage word vector and the question word vector.

Given the alignment matrix element as weights, we compute the new passage representation $h_i^t$ for the $t$-th layer by using weighted average over all the independent question representation $v^Q$ calculated from Eq 1, as shown in the following.

$$h_i^t = \sum_k^m \mathcal{A}_{ik}^t \cdot v_k^Q \tag{5}$$

where $h_i^{t-1} \in \mathbb{R}^{2d}$. Note the independent representation $v_k^Q$ for $k$-th question word is different with the shared weight question representation $u_k^Q$.

### 2.2.1 OUTER FUSION LAYERS

For each question-passage attention layer, its output of $h_i^t$, where $t \in N$, is concatenated to form the final output vector to represent the $i$-th passage word $C_i^0 = [h_i^1; \ldots; h_i^N]$. Increasing the number of layers $N$ allows an increasingly more complex representation for a passage word.

In order to regulate the flow of $N$ question-passage attention layers and to prevent the over-fitting problem, we use fusion layers, which is highway networks (Srivastava et al., 2015) using of GRU-like gating units and taking $C_i^0$ as its input:

$$\tilde{C}_i^t = \text{ReLU}(W_C^t \cdot C_i^{t-1} + b_C^t) \tag{6}$$

$$z^t = \sigma(W_z^t \cdot C_i^{t-1} + b_z^t) \tag{7}$$

$$C_i^t = (1 - z^t) \circ C^{t-1} + z^t \circ \tilde{C}_i^t \tag{8}$$

where $t \in K$, $K$ is the number of fusion layers, $W_C^t$, $W_z^t$ are the weights, $b_C^t$, $b_z^t$ are the bias of $t$-th fusion layer, and the transform gate $z^t$ is a non-linear activation function. The final result of fusion layers $C_i^N \in \mathbb{R}^{2Nd}$ is sent to self-attention models as input for processing.

## 2.3 Self-Attention Layers

Following the question-passage attention layers, self-attention layers propagate evidence through the passage context. This process is similar in spirit to the steps of exploring similarity or redundancy between answer candidates (e.g., "J.F.K" and "Kennedy" can, in fact, be equivalent despite their different surface forms) that have been shown to be very effective during answer merging stage (Ferrucci et al., 2010). More generally, propagating evidence among the passage words allows correct answers to have better evidence for the question than the rest part of the passage.

For a single self-attention layer, we first compute a self alignment matrix $\mathcal{S}^t \in \mathbb{R}^{n \times n}$ by comparing the passage representation itself,

$$\mathcal{S}^t(i, j) = \frac{\exp(h_i^{t-1} \cdot h_j^{t-1})}{\sum_k \exp(h_i^{t-1} \cdot h_k^{t-1})} \tag{9}$$

where $h_i^{t-1}$ is the $i$-th passage word as input for the $t$-th self-attention layer, initial value $h_i^0$ is defined as the final fused result $C_i^N$ from question-passage attention model in section 2.2.1.

Given the alignment matrix element as weights, evidences are propagate from the previous layer to the next to produce the new passage representation $h_i^t$ by using the weighted average over all the $t-1$ layer passage representation:

$$B_i^t = \sum_k^n \mathcal{S}_{ik}^t \cdot h_k^{t-1} \tag{10}$$

where $h_k^{t-1}$ is the passage representation for the $k$-th word at the $t-1$ self-attention layer, $B_i^t \in \mathbb{R}^{2Nd}$ is the output the self-attention layer and it will be sent to a fusion layer, described in section 2.3.1, to obtain the $t$-th layer passage representation $h_i^t$.

### 2.3.1 Inner Fusion Layers

To efficiently propagate evidence through the passage, we refine the self-attended representations by using multiple layers. At the end of each self-attention layer, a GRU-like gating mechanism (Hu et al., 2017) is used to decide what information to store and send to the next self-attention layer, by merging the newly produced representation of the current layer and the input representation from the previous layer,

$$\tilde{B}_i^t = \tanh(W_B^t \cdot [B_i^t; B_i^{t-1}; B_i^t \circ B_i^{t-1}] + b_B^t) \tag{11}$$

$$f^t = \sigma(W_f^t \cdot [B_i^t; B_i^{t-1}; B_i^t \circ B_i^{t-1}] + b_f^t) \tag{12}$$

$$h_i^t = (1 - f^t) \circ h^{t-1} + f^t \circ \tilde{B}_i^t \tag{13}$$

where $W_B^t$, $W_f^t$ are the weights, $b_B^t$, $b_f^t$ are the bias of $t$-th fusion layer, and $f^t$ is a non-linear activation function. The output $h_i^t$, whose dimensions are the same as its input vector $B_i^t$, is then sent to the next layer of self-attention model as input to calculate Eq 9 and Eq 10.

## 2.4 Output Layers

We directly follow Hu et al. (2017) and use a memory-based answer pointer networks to predict boundary of the answer. The memory-based answer pointer network contains multiple hops. For the $t$-th hop, the pointer network produces the probability distribution of the start index $p_s^t$ and the end index $p_e^t$ using a pointer network (Vinyals et al., 2015) respectively. If the $t$-th hop is not the last hop, then the hidden states for the start and end indices are transformed and fed into the next-hop prediction. The training loss is defined as the sum of the negative log probabilities of the last hop start and end indices averaged over all examples.

## 3 Experiments and Analysis

This paper focuses on the Stanford Question Answering Dataset (SQuAD) (Rajpurkar et al., 2016) to train and evaluate our model. SQuAD, which has gained a significant attention recently, is a

Table 2: Performance comparison of single models on the development set. Each setting contains five runs trained consecutively. Standard deviations across five runs are shown in the parenthesis for single models. Daggers indicate the level of significance.

| Attention Models | EM | | F1 | |
|---|---|---|---|---|
| | Max | Mean ($\pm$SD) | Max | Mean ($\pm$SD) |
| Iterative Aligner | 70.95 | 70.64 ($\pm$0.34) | 80.46 | 80.23 ($\pm$0.16) |
| Iterative Aligner, QPAtt+ | 71.21 | 71.11 ($\pm$0.31)† | 80.73 | 80.52 ($\pm$0.16) † |
| PhaseCond | 71.36 | 71.07 ($\pm$0.28) † | 80.76 | 80.53 ($\pm$0.22) † |
| PhaseCond, QPAtt+ | **71.85** | **71.60** ($\pm$0.22) ‡ | **81.13** | **81.04** ($\pm$0.17) ‡ |

largescale dataset consisting of more than 100,000 questions manually created through crowdsourcing on 536 Wikipedia articles. The dataset is randomly partitioned into a training set (80%), a development set (10%), and a blinded test set (10%). Two metrics are used to perform evaluation: Exact Match (EM) score which calculates the ratio of questions that are answered correctly by exact string match, and F1 score which calculates the harmonic mean of the precision and recall between predicted answers and ground true answers at the character level.

## 3.1 TRAINING DETAILS

Our input for the encoding layer in Section 2.1 includes a list of commonly used features. We use pre-trained GloVe 100-dimensional word vectors (Pennington et al., 2014), parts-of-speech tag features, named-entity tag feature, and binary features of exact matching (Chen et al., 2017) which indicate if a passage word can be exactly matched to any question word and vice versa. Following Hu et al. (2017), we also use question type (what, how, who, when, which, where, why, be, and other) features (Zhang et al., 2017) where each type is represented by a trainable embedding. We use CNN with 100 one-dimensional filters with width 5 to encode character level embedding. The hidden size is set as 128 for all the LSTM layers. Dropout (Srivastava et al., 2014) are used for all the learnable parameters with a ratio as 0.2. We use the Adam optimizer (Kingma & Ba, 2014) with an initial learning rate of 0.0006, which is halved when a bad checkpoint is met.

## 3.2 MAIN RESULTS OF MODEL COMPARISON

We compare our proposed model PhaseCond with a multi-layered attention model, the Iterative Aligner, as well as various other recently published systems, which include a single-layered model, BIDAF (Seo et al., 2017), and a single-layered model containing both the question-passage attention and self-attention, RNET (Wang et al., 2017). We first compare our proposed model PhaseCond with Iterative Aligner, which is employed by two top ranked systems MEMEN (Pan et al., 2017) and MReader (Hu et al., 2017) on the SQuAD leaderboard [1]. Since our goal is to show the effectiveness of our proposed model PhaseCond, we use a baseline system implementing MReader for the direct comparison. All the experiment settings are the same for PhaseCond and Iterative Aligner including the number of attention layers, input features, optimizer and learning rate, number of training steps and etc. As shown in Table 2 which summarizes the performance of single models, we achieve steady improvements when 1) additional question encoders are used to extend the passage-question attention function, denoted as QPAtt+, as detailed in Section 2.1 and Section 2.2, and 2) on top of that, using PhaseCond making our model better than using Iterative Aligner. Specifically, PhaseCond's computational path for two question-aware passage attention layers $\mathcal{L}^Q$ and two self-attention layers $\mathcal{L}^S$ goes from $\mathcal{L}_1^Q \rightarrow \mathcal{L}_2^Q \rightarrow \mathcal{F}_{outer} \rightarrow \mathcal{L}_1^S \rightarrow \mathcal{F}_{inner} \rightarrow \mathcal{L}_2^S \rightarrow \mathcal{F}_{inner}$. On the other hand, Iterative Aligner builds path in turn through different kinds of attention layers: $\mathcal{L}_1^Q \rightarrow \mathcal{F}_{inner} \rightarrow \mathcal{L}_1^S \rightarrow \mathcal{F}_{inner} \rightarrow \mathcal{L}_1^Q \rightarrow \mathcal{F}_{inner} \rightarrow \mathcal{L}_2^S \rightarrow \mathcal{F}_{inner}$.

As shown in Table 3, in the single model setting, our model PhaseCond is clearly more effective than all the single-layered models (BiDAF and RNET) and multi-layered models (MReader and Iterative Aligner). We draw the same conclusion for the ensemble model setting, despite that the RNET works better on the Dev EM measure. The EM result of our baseline Iterative Aligner is lower

---

[1]https://rajpurkar.github.io/SQuAD-explorer/

Table 3: The performance of our models and published results of competing attention-based archi-tectures. To perform a fair comparison as much as possible, we collect the results of BiDAF (Seo et al., 2017) and RNET (Wang et al., 2017) from their recently published papers instead of using the up-to-date performance scores posted on the SQuAD Leaderboard. Our directly available baseline is one implementation of MReader, re-named as Iterative Aligner which has very similar results as those of MReader (Hu et al., 2017) posted on the SQuAD Leaderboard on Jul 14, 2017.

| | Single Model | | Ensemble Models | |
| | Dev Set | Test Set | Dev Set | Test Set |
| **Attention-based Systems** | **EM / F1** | **EM / F1** | **EM / F1** | **EM / F1** |
|---|---|---|---|---|
| BiDAF (Seo et al., 2017) | 67.7 / 77.3 | 68.0 / 77.3 | 73.3 / 81.1 | 73.3 / 81.1 |
| RNET (Wang et al., 2017) | 71.1 / 79.5 | 71.3 / 79.7 | **75.6** / 82.8 | 75.9 / 82.9 |
| MReader (Hu et al., 2017) | N/A | 71.0 / 80.1 | N/A | 74.3 / 82.4 |
| Iterative Aligner (Hu et al., 2017) | 70.2 / 79.6 | N/A | 73.3 / 81.6 | N/A |
| PhaseCond | **72.1 / 81.4** | **72.6 / 81.4** | 74.8 / **83.3** | **76.1 / 84.0** |

Table 4: Varying number of question-passage attention layers and self-attention layers. We set layer number in PhaseCond for question-passage attention model (denoted as QPAtt) and self-attention model (denoted as SelfAtt) respectively. L1 means a single layer and L2 means two stacking layers.

| | EM | | F1 | |
| **Attention Layers** | **Max** | **Mean ($\pm$SD)** | **Max** | **Mean ($\pm$SD)** |
|---|---|---|---|---|
| QPAtt-L1, SelfAtt-L1 | 71.26 | 71.29 ($\pm$0.19) | 80.83 | 80.68 ($\pm$0.17) |
| QPAtt-L1, SelfAtt-L2 | **72.05** | 71.56 ($\pm$0.30) | 81.11 | 80.98 ($\pm$0.15) |
| QPAtt-L2, SelfAtt-L1 | 71.26 | 70.88 ($\pm$0.30) | 80.79 | 80.41 ($\pm$0.31) |
| QPAtt-L2, SelfAtt-L2 | 71.85 | **71.60 ($\pm$0.22)** | **81.13** | **81.04 ($\pm$0.17)** |

than RNET, confirming that the problem is not caused by our proposed model. Our explanations is that 1) RNET uses a different feature set (e.g., GloVe 300 dimensional word vectors are employed) and different encoding steps (e.g., three GRU layers are used for encoding question and passage representations), and 2) RNET uses a different ensemble method from our implementation.

### 3.3 ANALYSIS ON ATTENTION LAYERS

Table 4 shows the performance with different number of layers for both question-passage attention phase and self-attention phase. We change the layer number separately to compare the performance. For the question-passage attention phase, using single layer doesn't degrade the performance signifi-cantly from the default setting of two layers, resulting in a different conclusion from Hu et al. (2017); Xiong et al. (2017). Intuitively, this is largely expected because representing the passage repeatedly with the same question doesn't constantly add more information. In contrast, multiple stacking lay-ers are needed to allow the evidence fully propagated through the passage. This is exactly what we observed in two stacking layered self-attention phase.

In Figure 3, we visualize the attention matrices for each layer to show dynamic attention changes. The model is based on the main setting which has two question-passage layers and two self-attention layers. We observed several critical trends. First, the first layer of the question-passage attention phase can successfully align question keywords with the corresponding passage keywords, as shown in Figure 3a. For example, the question keyword "represented" have been successfully aligned with related passage keywords "champion", "defeated", and "earned". Second, patterns of striped color in Figure 3a indicate similar weights among all the passage words, meaning that it becomes indistinguishable among passage words, and therefore adding another layer of question-passage attention model degrades the alignment quality dramatically. This observation is meaningful which shows that repeatedly representing a passage word regarding the same question representation can make the passage embedding become closer to the original question representation. Third, when comparing Figure 3c and Figure 3d, we observed that the color is diluted for most of the weights in the second layer of self-attention phase, meanwhile a small portion of weights is strengthened,

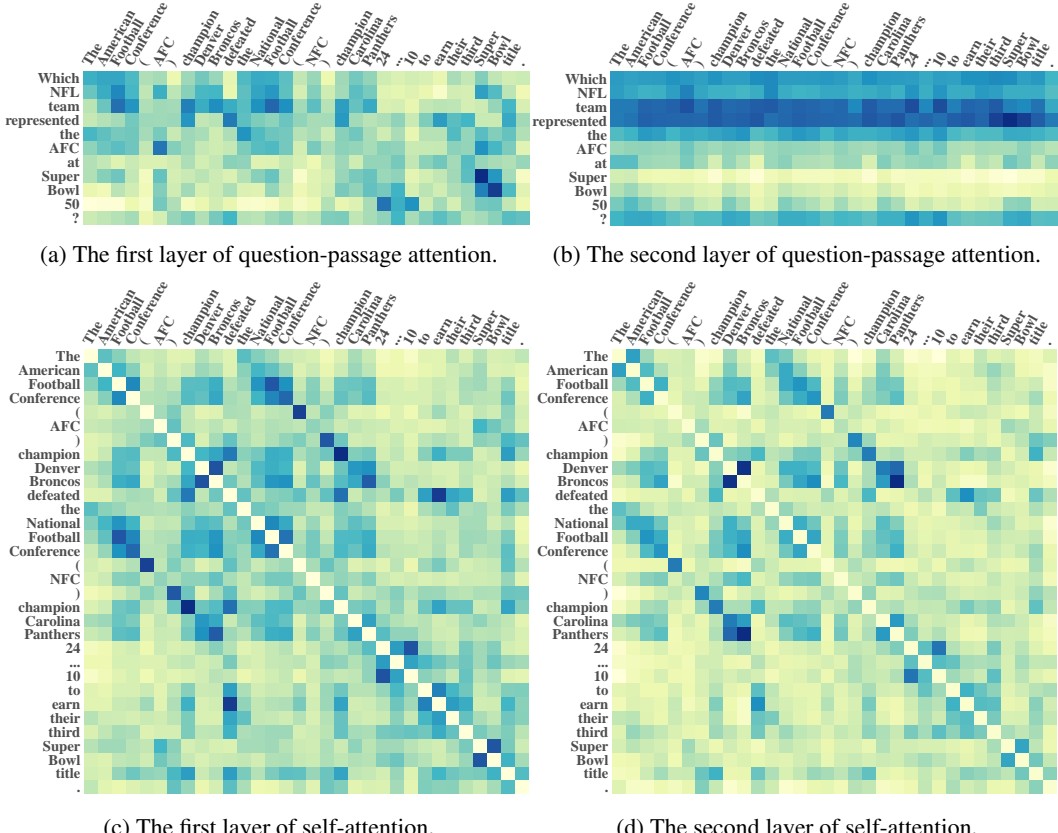

Figure 3: Dynamic attention changes of multiple layers on a visualized example. The matrices are the attention weights computed by the dot-product attention function before any normalization. Generally, the darker the color is the higher the weight is (the only exception is Figure 3b which contains negative values). Given the question "Which NFL team represented the AFC at Super Bowl 50?", the system correctly detects the answer "Denver Broncos" from the passage part "The American Football Conference (AFC) champion Denver Broncos defeated the National Football Conference (NFC) champion Carolina Panthers 2410 to earn their third Super Bowl title."

suggesting that information propagation is converging. For example, in Figure 3d as the last attention layer, the phrase "Denver Broncos" becomes more concentrated on the phrase "Carolina Panthers" than that of Figure 3c. In contrast, "Denver Broncos" becomes less focused on the other keywords (e.g., "champion" and "title") of the same passage.

## 4 CONCLUSION

In this paper, we introduce a general framework PhaseCond, on multi-layered attention models with two phases including a question-aware passage representation phase and an evidence propagation phase. The question-aware passage representation phase has a stack of question-aware passage attention models, followed by outer fusion layers that regularize concatenated passage representations. The evidence propagation phase has a stack of self-attention layers, each of which is followed by inner fusion layers that control the information to propagate and output. Also, an improved attention mechanism for PhaseCond is proposed based on a popular dot-product attention function by simultaneously encoding both the independent question embedding layers, the weight-sharing question embedding layer and weight-sharing passage embedding layer. The experimental results show that our model significantly outperforms single-layered or multiple-layered attention networks on blinded test data of SQuAD. Moreover, our in-depth quantitative analysis and visualizations provide meaningful findings for both question-aware passage attention mechanism and self-matching attention mechanism.

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
