# OpenReview forum: "Phase Conductor on Multi-layered Attentions for Machine Comprehension"
_ICLR.cc/2018/Conference — Reject_

### Official Review · AnonReviewer2 · 2017-11-19
**Improvements on SQuAD, but overall weak**

**Rating:** 5
**Confidence:** 5

**Review:**

Summary: The paper introduces "Phase Conductor", which consists of two phases, context-question attention phase and context-context (self) attention phase. Each phase has multiple layers of attention, for which the paper uses a novel way to fuse the layers, and context-question attention uses different question embedding for getting the attention weight and getting the attention vector. The paper shows that the model achieves state of the art on SQuAD among published papers, and also quantitatively and visually demonstrates that having multiple layers of attention is helpful for context-context attention, while it is not so helpful for context-question attention.


Note: While I will mostly try to ignore recently archived, non-published papers when evaluating this paper, I would like to mention that the paper's ensemble model currently stands 11th on SQuAD leaderboard.


Pros:
- The model achieves SOTA on SQuAD among published papers.
- The sequential fusing (GRU-like) of the multiple layers of attention is interesting and novel. Visual analysis of the attention map is convincing.
- The paper is overall well-written and clear.

Cons:
- Using different embedding for computing attention weights and getting attended vector is not entirely novel but rather an expected practice for many memory-based models, and should cite relevant papers. For instance, Memory Networks [1] uses different embedding for key (computing attention weight) and value (computing attended vector).
- While ablations for number of attention layers (1 or 2) were visually convincing, numerically there is a very small difference even for selfAtt. For instance, in Table 4, having two layers of selfAtt (with two layers of question-passage) only increases max F1 by 0.34, where the standard deviation is 0.31 for the one layer. While this may be statistically significant, it is a very small gain nonetheless.
- Given the above two cons, the main contribution of the paper is 1.1% improvement over previous state of the art. I think this is a valuable engineering contribution, but I feel that it is not well-suited / sufficient for ICLR audience.


Questions:
- page 7 first para: why have you not tried GloVe 300D, if you think it is a critical factor?


Errors:
- page 2 last para: "gives an concrete" -> "gives a concrete"
- page 2 last para: "matching" -> "matched"
Figure 1: I think "passage embedding h" and "question embedding v" boxes should be switched.
- page 7 3.3 first para: "evidence fully" -> "evidence to be fully"


[1] Jason Weston, Sumit Chopra, and Antoine Bordes. Memory Networks. ICLR 2015.

---

> ### Author Response · Authors · 2018-01-05
> **Contribution emphasized, new experiment conducted with better performance gain.**
>
> We thank the reviewer for acknowledging the contributions of our work and for providing insightful comments. Several issues on the experiments have been fixed during the review period, and we see the significant performance gain on both dev and test dataset. We have responded each of the comments below.
>
> Comment 1.
> Original Comment:
> Using different embedding for computing attention weights and getting attended vector is not entirely novel but rather an expected practice for many memory-based models, and should cite relevant papers. For instance, Memory Networks [1] uses different embedding for the key (computing attention weight) and value (computing attended vector).
>
> Response:
> The reviewer is right that our research is not the first one to explore the idea of using different embeddings. Although the motivation and approaches we use are very different from what has been proposed in Memory Network paper (e.g., we explicitly added a separate question embedding for dot-product attention function in order to improve attention model’s robustness), we do believe that it is a good idea to incorporate the memory network paper into our citations. We will update the citation in the final version of the paper. However, we do want to emphasize that our proposed structure of PhaseCond is novel and quite effective on the machine comprehension task for the following reasons. First, our network consists of multiple phases, 1) each phase has a stack of attention (or functional) layers, 2) each layer is followed by a stack of inner, and 3) at the end of each phase outer fusion layers are configured. Second, we use a novel approach to increase the attention model’s robustness with respect to the dot-product attention function by adding a separate question/query embedding. Overall our proposed PhaseCond model is significantly more effective compared with existing attention-based architectures.
>
>
> Comment 2:
> Original Comment:
> While ablations for number of attention layers (1 or 2) were visually convincing, numerically there is a very small difference even for selfAtt. For instance, in Table 4, having two layers of selfAtt (with two layers of question-passage) only increases max F1 by 0.34, where the standard deviation is 0.31 for the one layer. While this may be statistically significant, it is a very small gain nonetheless.
>
> Response:
> Our updated ablation study shows (please see our paper and replies to reviewer 3) that even the well-developed approaches (e.g., highway network, or residual connections) have a small gain on SQuAD dataset, which means that the dataset itself is very challenging. Considering the fact that the higher F1 score is, the more difficult to improve the performance, we believe that our improvements over the state-of-art multi-layered attention model are not trivial.
>
> Comment 3:
> Original Comment:
> Given the above two cons, the main contribution of the paper is 1.1% improvement over the previous state of the art. I think this is a valuable engineering contribution, but I feel that it is not well-suited / sufficient for ICLR audience.
>
> Response:
> We have conducted experiments again after fixing several bugs and our single model version has managed to achieve 74.405 on exact match (compares to 73.240 reported in the manuscript to review) and 82.742 on F1 score (compares to 81.933 reported in the manuscript to review) on the test dataset. Compared to MReader, which is the best baseline reported in the paper with EM 71 and F1 80.1, our model delivers a significant improvement by 2-3%. We believe that this reflects algorithmic innovations rather than engineering contributions.
>
> Comment 4:
> Original Comment:
> page 7 first para: why have you not tried GloVe 300D, if you think it is a critical factor?
>
> Response:
> We have incorporated GloVe 300D in our latest experiments. Along with other bug fixes, we have noticed a 1% performance boost over the results reported in the previous version of the paper.

---

### Official Review · AnonReviewer3 · 2017-11-21
**New model architecture with marginal improvement, more ablation tests and analysis may help**

**Rating:** 5
**Confidence:** 4

**Review:**

This paper proposes a new machine comprehension model, which integrates several contributions like different embeddings for gate function and passage representation function, self-attention layers and highway network based fusion layers. The proposed method was evaluated on the SQuAD dataset only, and marginal improvement was observed compared to the baselines.

(1) One concern I have for this paper is about the evaluation. The paper only evaluates the proposed method on the SQuAD data with systems submitted in July 2017, and the improvement is not very large. As a result, the results are not suggesting significance or generalizability of the proposed method.

(2) The paper gives some ablation tests like reducing the number of layers and removing the gate-specific question embedding, which help a lot for understanding how the proposed methods contribute to the improvement. However, the results show that the deeper self-attention layers are indeed useful (but still not improving a lot, about 0.7-0.8%). The other proposed components contribute less significant. As a result, I suggest the authors add more ablation tests regarding (1) replacing the outer-fusion with simple concatenation (it should work for two attention layers); (2) removing the inner-fusion layer and only use the final layer's output, and using residual connections (like many NLP papers did) instead of the more complicated GRU stuff.

(3) Regarding the ablation in Table 2, my first concern is that the improvement seems small (~0.5%). As a result, I am wondering whether this separated question embedding really brings new information, or the similar improvement can be achieved by increasing the size of LSTM layers. For example, if we use the single shared question embeddings, but increase the size from 128 to some larger number like 192, can we observe similar improvement. I suggest the authors try this experiment as well and I hope the answer is no, as separated input embeddings for gate functions was verified to be useful in some "old" works with syntactic features as gate values, like "Semantic frame identification with distributed word representations" and "Learning composition models for phrase embeddings" etc.

(4) Please specify which version of the SQuAD leaderboard is used in Table 3. Is it a snapshot of the Jul 14 one? Because this paper is not comparing to the state-of-the-art, no specification of the leaderboard version may confuse the other reviewers and readers. By the way, it will be better to compare to the snapshot of Oct 2017 as well, indicating the position of this work during the submission deadline.

Minor issues:

(1) There are typos in Figure 1 regarding the notations of Question Features and Passage Features.

(2) In Figure 1, I suggest adding an "N \times" symbol to the left of the Q-P Attention Layer and remove the current list of such layers, in order to be consistent to the other parts of the figure.

(3) What is the relation between the "PhaseCond, QPAtt+" in Table 2 and the "PhaseCond" in Table 3? I was assuming that those are the same system but did not see the numbers match each other.

---

> ### Author Response · Authors · 2018-01-05
> **Contribution emphasized, ablation tests performed with details, and new experiment conducted with better performance gain**
>
> We thank the reviewer for acknowledging the contributions of our work and for providing insightful comments. Several issues on the experiments are fixed during the review period, and we see the significant performance gain on both dev and test dataset compares to the results reported in the previous version of the paper. To address concerns from the reviewer especially in the section of the ablation test, we have also conducted additional experiments. We have responded each of the comments below.
>
> Comment 1.
> Original Comment:
> One concern I have for this paper is about the evaluation. The paper only evaluates the proposed method on the SQuAD data with systems submitted in July 2017, and the improvement is not very large. As a result, the results are not suggesting significance or generalizability of the proposed method.
>
> Response:
> We have conducted experiments again after several bug fixes and our single model version has managed to achieve 74.405 on exact match (compares to 73.240 reported in the draft to review) and 82.742 on F1 score (compares to 81.933 reported in the draft to review) in the test dataset. Compares to MReader, which is the best baseline reported in the paper with EM 71 and F1 80.1, our model delivered significant improvement with 2-3% performance improvement. We believe that the updated results reflected a significant improvement by applying our model. The standard deviation across multiple models is much lower (around 0.1 for our model and 0.5 for strongest baseline). We will update those results in the final version of the paper.
>
> Although we only evaluate our method on SQuAD, we note that our approach is a general method to handle multi-layered attentions and can be applied to any application that matches a query and a context (e.g., VQA, image or text search) that is not limited to SQuAD. We do agree with the reviewer that this is an important aspect of the paper. We will add discussions of the generalizability in the final version of the paper.
>
> Comment 2:
> Original Comment:
> The paper gives some ablation tests like reducing the number of layers and removing the gate-specific question embedding, which help a lot for understanding how the proposed methods contribute to the improvement. However, the results show that the deeper self-attention layers are indeed useful (but still not improving a lot, about 0.7-0.8%). The other proposed components contribute less significant. As a result, I suggest the authors add more ablation tests regarding (1) replacing the outer-fusion with simple concatenation (it should work for two attention layers); (2) removing the inner-fusion layer and only use the final layer's output, and using residual connections (like many NLP papers did) instead of the more complicated GRU stuff.
>
> Response:
> The reviewer made a good point that we should add more ablation tests and we have presented the results below. Those results are after several bug fixes and they, in general, perform much better than models previously reported in the draft.
>
> Each setting has 3 runs without any model selection:
> 1)	The proposed approach (PhaseCond)
>               F1: 81.16 (SD: 0.10) | EM: 71.99 (SD: 0.17)
> remark: After fixing some bugs in our model, we observe that the proposed model outperforms all the comparable baselines and the variance of the models are the lowest (SD stands for standard deviation).
> 2)	replacing the outer-fusion with simple concatenation (PhaseCond-Outer+Concat)
>               F1: 76.36 (SD: 0.20) | EM: 66.21 (SD: 0.43)
> remark: The result shows that 1) our outer layer structure is critical to the model and 2) outer layer cannot be simply replaced by concatenation.
> 3)	removing the inner-fusion layer and only use the final layer's output (PhaseCond-Inner)
>               F1: 80.32 (SD: 0.31) | EM: 70.91 (SD: 0.72)
> remark: It demonstrates that the inner layer implemented by highway network (RK Srivastava et al., ‎2015) can be complementary to LSTM-style outer layer and it is not replaceable.
> 4)	using residual connections (PhaseCond-Inner+Residual)
>               F1: 80.47 (SD: 0.19) | EM: 71.14 (SD: 0.26)
> remark: Our result shows that adding residual layers is a little bit helpful but cannot replace either inner layer or outer layer of the proposed PhaseCond. The residual connections are designed for deep networks with hundreds of layers, but our model is attention-based and comparably shallow.
>
> Note: all the correctness can be directly verified via changing our latest experiment code (all the sources, data are available):
> https://worksheets.codalab.org/bundles/0xfc1bc55358b049029514f1018ff70ece/

---

> ### Author Response · Authors · 2018-01-05
> **The second part of authors' reply**
>
> Comment 3:
> Original Comment:
> Regarding the ablation in Table 2, my first concern is that the improvement seems small (~0.5%). As a result, I am wondering whether this separated question embedding really brings new information, or the similar improvement can be achieved by increasing the size of LSTM layers. For example, if we use the single shared question embeddings, but increase the size from 128 to some larger number like 192, can we observe similar improvement. I suggest the authors try this experiment as well and I hope the answer is no, as separated input embeddings for gate functions was verified to be useful in some "old" works with syntactic features as gate values, like "Semantic frame identification with distributed word representations" and "Learning composition models for phrase embeddings" etc.
>
> Response:
> We thank the reviewer for the insightful comments. We have conducted additional experiments by changing the number of units.
> 1)	PhaseCond (use the best hidden size 150)
>               F1: 81.16 (SD: 0.10) | EM: 71.99 (SD: 0.17)
>               We find that increasing the hidden size (original 128) can improve the performance, and the optimal parameter we use now is 150.
> 2)	Increase the hidden size to 192
>               F1: 80.68 (SD: 0.90) | EM: 71.29 (SD: 1.07)
>               Further increasing the hidden size can cause the performance drop and hurt the stability.
> 3)	Remove the separated question embedding from PhaseCond
>               F1: 80.58 (SD: 0.47) | EM: 71.16 (SD: 0.24)
>               There are reasons for us to train a separated question embedding:  1) for similarity between question and passage, they have to be in the same “embedding space” and therefore we jointly train them, but 2) the attention model is to use the question to represent the passage, so for the question embedding, there’s no need to consider the passage itself. Training question and passage parameters jointly may not be suited for question embeddings to represent a passage, so we train a separate question embedding to “stabilize” the quality of our attention model. It is also optimal for the dot-product attention function we chose, which doesn’t have the parameter W (it consumes too much VRAM that we cannot afford).
>
> Note: all the correctness can be directly verified via changing our latest experiment code (all the sources, data are available):
> https://worksheets.codalab.org/bundles/0xfc1bc55358b049029514f1018ff70ece/
>
> Comment 4:
> Original Comment:
> Please specify which version of the SQuAD leaderboard is used in Table 3. Is it a snapshot of the Jul 14 one? Because this paper is not comparing to the state-of-the-art, no specification of the leaderboard version may confuse the other reviewers and readers. By the way, it will be better to compare to the snapshot of Oct 2017 as well, indicating the position of this work during the submission deadline.
>
> Response:
> We could have compared the performance of our papers to those on the leaderboard. However, in reality, we noticed that at the time when we were writing the paper, most of the systems on the top of the leaderboard do not have publications released. Even for those systems that have publications, their performance on the leaderboard are constantly changing and there is no guarantee that their systems are implemented strictly according to their original paper. For those reasons, we chose to compare results on the published papers. During the time we developed our model (around Aug - Sep in 2017), our ranking was among top 5 on SQuAD. After we submitted the paper, our ranking is as good as top 10 (around Oct 2017).

---

### Official Review · AnonReviewer1 · 2017-11-29
**Competent work.  This paper seems to represent a legitimate player in the SQuAD leaderboard game.**

**Rating:** 8
**Confidence:** 3

**Review:**

This paper introduces a fairly elaborate model for reading comprehension evaluated on the SQuAD dataset.   The model is shown to improve on the published results but not as-of-submission leaderboard numbers.

The main weakness of the paper in my opinion is that the innovations seem to be incremental and not based on any overarching insight or general principle.  A less significant issue is that the English is often disfluent.

Specific comments: I would remove the significance daggers from table 2 as the standard deviations are already reported and the null hypothesis for which significance is measured seems unclear.  I am also concerned to see test performance significantly better than development performance in table 3.  Other systems seem to have development and test performance closer together.  Have the authors been evaluating many times on the test data?

---

> ### Author Response · Authors · 2018-01-05
> **Contribution emphasized, new experiment conducted with better performance gain.**
>
> We thank the reviewer for acknowledging the contributions of our work and for providing insightful comments. Several issues on the experiments are fixed during the review period, and we see the significant performance gain on both dev and test dataset. We have responded each of the comments below.
>
> Comment 1.
> Original Comment:
> The main weakness of the paper in my opinion is that the innovations seem to be incremental and not based on any overarching insight or general principle.  A less significant issue is that the English is often disfluent.
> Response:
> Thanks for your comments and feedbacks. We have conducted experiments again after fixing several bugs and our single model version has managed to achieve 74.405 on exact match (compared to 73.240 reported in the manuscript to review) and 82.742 on F1 score (compared to 81.933 reported in the draft to review) regarding the test dataset. Compared to MReader, which is the best baseline reported in the paper with EM 71 and F1 80.1, we believe that our model delivered significant improvement by 2-3%.
>
> When we were building the mode, we did have a principle in mind which was to divide the information flow into two phases: Question-Paragraph attention phase and self-attention phase, each equipped with a multi-layered attention structure. We will detail the rationales of such a design and how it is motivated by real examples in the final version of the paper.
>
> The reviewer also made a remark in the language of the paper. We will correct all the result and grammatical mistakes and have professionals to proofread the paper when we are preparing the final version of the paper.
>
>
> Comment 2:
> Original Comment:
> Specific comments: I would remove the significance daggers from table 2 as the standard deviations are already reported and the null hypothesis for which significance is measured seems unclear.  I am also concerned to see test performance significantly better than development performance in table 3.  Other systems seem to have development and test performance closer together.  Have the authors been evaluating many times on the test data?
> Response:
> We fixed some bugs in our last experiments (e.g., one outer fusion layer is missing, hidden size 128 is not optimal, etc) and observed much stronger performance. As we have mentioned in the previous paragraph, the performance improvement is more than 1% over the last results reported in the manuscript to review.
>
> It is a well-known fact in the community that the test dataset yields slightly better performance than the development set. This has been reported on the SQuAD dataset (e.g., RNET (Wang et al., 2017), BIDAF (Seo et al., 2017)). We have worked on the SQuAD dataset for a long time and we are pretty confident that this is the expected behavior on SQuAD.

---

### Decision · Program_Chairs · 2018-01-29
**ICLR 2018 Conference Acceptance Decision**

**Decision:**

Reject

**Comment:**

Generally solid engineering work but a bit lacking in terms of novelty and some issues with clarity. At the end of the day the empirical gains are not sufficient for acceptance - the results are state-of-the-art relative to published work, but not in the top 10 based on the official leaderboard (not even at time of submission). Since the technical contributions are small and the engineering contributions have been made obsolete by concurrent work, I suggest rejection.